

# Prevalence and extent of chronic periodontitis and its risk factors in a Portuguese subpopulation: a retrospective cross-sectional study and analysis of Clinical Attachment Loss

Vanessa Machado[1,2,*], João Botelho[1,2,*], António Amaral[2], Luís Proença[2], Ricardo Alves[1,2], João Rua[2], Maria Alzira Cavacas[3], Ana Sintra Delgado[2] and José João Mendes[2]

[1] Department of Periodontology, Clinical Research Unit, Centro de Investigação Interdisciplinar Egas Moniz (CiiEM), Instituto Universitário Egas Moniz, Almada, Portugal
[2] Clinical Research Unit, Centro de Investigação Interdisciplinar Egas Moniz (CiiEM), Instituto Universitário Egas Moniz, Almada, Portugal
[3] Environmental Health, Centro de Investigação Interdisciplinar Egas Moniz (CiiEM), Instituto Universitário Egas Moniz, Almada, Portugal
* These authors contributed equally to this work.

Corresponding author
João Botelho,
jbotelho@egasmoniz.edu.pt,
joaobotelho09@gmail.com

## ABSTRACT

**Objectives.** To assess the prevalence and extent of chronic periodontitis, and its risk factors in a Portuguese subpopulation referred to periodontal examination.

**Methods.** This retrospective cross-sectional study used a subset of data from patients who sought dental treatment in a university dental clinic in the Lisbon metropolitan area. The sample consisted of 405 individuals (225 females/180 males), aged 20–90 years. All patients underwent a full-mouth periodontal examination and chronic periodontitis was defined as Clinical Attachment Loss (CAL) $\geq$ 3 mm affecting two or more teeth. Aggressive periodontitis cases were excluded from the analysis.

**Results.** Prevalence of chronic periodontitis was 83.5% (95% CI [80.4–86.6%]). For these subjects, CAL $\geq$ 3 mm affected 86.0% (95% CI [84.7–87.2]) of sites and 83.7% (95% CI [81.7–85.6]) of teeth, respectively. Mean CAL ranged from 3.6 to 4.3 mm, according to age. In the multivariate logistic regression model, smoking (OR = 3.55, 95% CI [1.80–7.02]) and older age (OR = 8.70, 95% CI [3.66–20.69] and OR = 4.85, 95% CI [2.57–9.16]), for 65+ and 45–64 years old, respectively, were identified as risk indicators for CAL $\geq$ 3 mm.

**Conclusions.** This particular Portuguese adult subpopulation had a high prevalence of chronic periodontitis, with severe and generalized clinical attachment loss, and its presence was significantly associated with age and smoking. This data should serve to prepare future detailed epidemiological studies and appropriate public health programs.

## INTRODUCTION

Chronic periodontitis is an inflammatory disease characterized by a polymicrobial breakdown of host homeostasis and a progressive destruction of tooth-supporting structures (*Pihlstrom, Michalowicz & Johnson, 2005*; *Darveau, 2010*), and its epidemiology and risk factors have been broadly studied (*Albandar, 2002*; *Albandar & Rams, 2002*; *Brett et al., 2005*; *Burt, 2005*).

Periodontal diseases have a significant impact on oral health-related quality of life, especially with the worsening and extension of the disease in which it presents higher destructive consequences (*Buset et al., 2016*). There are important risk factors/indicators for periodontal disease such as alcohol (*Wang et al., 2016*), overweight and obesity (*Keller et al., 2015*), smoking (*Burt, 2005*) and diabetes (*Preshaw et al., 2012*). Also, periodontitis can be a risk factor for several systemic diseases (*Preshaw et al., 2012*; *Bahekar et al., 2007*; *Humphrey et al., 2008*; *Nibali et al., 2013*; *Lafon et al., 2014*; *Leira et al., 2017*; *Fuggle et al., 2016*; *Maisonneuve, Amar & Lowenfels, 2017*; *Papageorgiou et al., 2017*).

Some European epidemiological studies have demonstrated the high prevalence of periodontitis among the populations (*Bouchard et al., 2006*; *Bourgeois, Bouchard & Epidemiology, 2007*; *Aimetti et al., 2015*; *Schutzhold et al., 2015*; *Holde et al., 2017*). However, data on the prevalence and risk factors for periodontal disease in the Portuguese population are still missing. According to the latest Portuguese Oral National Health Survey, the prevalence of periodontitis was 10.8% in adults and 15.3% in the elderly (*DGS, 2015*). This nationwide survey used the Community Periodontal Index (CPI), with its recognized limitations. To the best of our knowledge, there are no epidemiological studies that used full-mouth periodontal examination (FMEP) methodology to estimate the prevalence of periodontitis regarding Portuguese samples.

The aim of this study was to assess the prevalence, severity, and extent of chronic periodontitis through full-mouth examination of CAL, and its association with sociodemographic, behavioral and environmental risk factors, in a Portuguese adult subpopulation, of a suburban area of the Lisbon Region, forwarded to periodontal examination.

## MATERIAL AND METHODS

The study was conducted in accordance with the Declaration of Helsinki of 1975, as revised in 2013, and approved by the Ethics Committee of Egas Moniz (Ethical Application Ref: 595). Written informed consent was obtained from all participants during the first appointment. After the examination, the participants were informed of their periodontal status, and those with diagnosed periodontal diseases were advised to follow the proper treatment. This protocol followed the STrengthening the Reporting of OBservational studies in Epidemiology (STROBE) guidelines (*Von Elm et al., 2014*).

### Study subjects

All participants were patients of Egas Moniz Dental Clinic (Almada, Portugal). This university clinic, located in the municipality of Almada, in Setúbal Peninsula (a NUTS III

subregion, part of NUTS II Lisbon Region), provides dental health services to the general public.

At the first appointment, patients were submitted to a dental triage protocol, with the application of a self-reported health questionnaire and oral and dental examinations, to guide their treatment needs. Regarding periodontal triage, patients were assessed using the Periodontal Screening and Recording (PSR) procedure (*Landry & Jean, 2002*), and, if diagnosed with code 2 or higher, they were forwarded to a periodontology appointment.

## Patient selection

This retrospective cross-sectional study analyzed patients who attended the dental clinic between September 2015 and March 2017. From a total of 3,648 subjects who sought the first consultation in the university dental clinic during that period, 1,501 (41%) patients were referred to the periodontology department, based on their triage status. From these, 459 attended a periodontal consultation and were considered for this study. Fifty-two participants were excluded due to incomplete questionnaires and periodontal data, and two subjects diagnosed with aggressive periodontitis. Hence, a final sample size of 405 subjects was obtained (11% of the total, 27% of the patients forwarded for periodontal treatment).

## Health questionnaire

Before clinical examinations, all patients answered a general and oral health questionnaire that included information such as age, gender, educational level, employment status, general medical history and medication, smoking status and oral hygiene habits.

## Clinical data
## Periodontal status

Five well-trained and calibrated periodontists (RA, JC, CI, FJ, LA) performed all dental and periodontal examinations. Periodontal examinations were performed using CDC/AAP full-mouth periodontal examination (FMEP) methodology (*Eke et al., 2012b*). We defined chronic periodontitis as CAL $\geq$ 3 mm affecting two or more teeth (*Susin et al., 2011*). All permanent fully erupted teeth were examined, excluding third molars, retained roots, and implants. The evaluated parameters were: missing teeth, presence or absence of supragingival biofilm (SB), probing depth (PD), bleeding on probing (BOP), gingival recession (REC) and clinical attachment loss (CAL). SB and BOP were scored on four surfaces of each tooth (mesial, distal, buccal and lingual). At six sites per tooth (mesiobuccal, mid-buccal, distobuccal, mesiolingual, mid-lingual and distolingual), PD was measured as the distance from the cementoenamel junction (CEJ) to the bottom of the pocket and REC as the distance from the CEJ to the free gingival margin, and this assessment was assigned a negative sign if the gingival margin was located coronally to the CEJ. CAL was calculated as the algebraic sum of PD and REC. It was used a CP-12 SE (Hu-Friedy, Chicago, IL, USA).

## Measurement reproducibility

Prior to the initiation of the study, all examiners were submitted to theoretical and practical training in a total of ten volunteer non-study patients suffering from moderate to severe

periodontitis. The inter-examiner correlation coefficients, at subject level, ranged from 0.76 to 0.97 and between 0.91 and 0.99, for mean PD and mean CAL, respectively.

## Covariates

Sociodemographic variables and several periodontal disease risk factors were selected as confounding variables. The selected variables were: age, gender, educational level, employment status, smoking status, Body Mass Index (BMI), time elapsed since the last dental appointment, consultation motive and oral hygiene habits.

Educational level was assessed as three categories: elementary (1–4 years), middle (5–12 years) and higher (>12 years) education. Employment status of each participant was classified as: employed, unemployed or retired. Smoking status was defined as non-smoker or smoker. Active smokers were further divided into three categories: light smokers (<10 cigarettes per day), medium smokers (10–20), heavy smokers (>20). The height of the participants was measured in centimeters, using a hard ruler installed vertically and secured with a stable base. Weight was assessed in kilograms using mechanical scales. BMI was calculated as the ratio of the individual's body weight to the square of their height. Four BMI categories were defined using WHO criteria (*Kopelman, 2000*): underweight (BMI $< 18.5$ kg/m$^2$), normal weight (BMI 18.5–24.9 kg/m$^2$), overweight (BMI 25–29.9 kg/m$^2$) and obese (BMI $\geq 30$ kg/m$^2$). The time elapsed since last dental consult was classified into five categories (never visited, less than one year, 1–2 years, 3–4 years, 5 years or over). Consultation motives were classified as routine, aesthetics, pain, functional or other. Oral hygiene habits were assessed by information on toothbrush frequency (2–3 times/daily, one time daily, 2–6 times/weekly) and dental floss use.

## Data analysis

Data analysis was performed using IBM SPSS Statistics version 24.0 for Windows (IBM Corp., Armonk, NY, USA). Descriptive and inferential statistics methodologies were applied. In the latter, Mann–Whitney and Kruskal-Wallis tests were used to compare the clinical data as a function of the sociodemographic variables. Further, logistic regression analysis was used to model the relationship between chronic periodontitis and several risk indicators. Preliminary analyses were performed using univariate models. Next, a multivariate model was constructed for the outcome variable CAL $\geq 3$ mm. Only variables showing a significance $p \leq 0.25$ in the univariate model were included in the multivariate stepwise procedure. Predictor variables considered in this procedure were: age (years), smoking status, education (years), employment status, last dental visit and dental floss use. The contribution of each variable to the model was evaluated by Wald statistics. Interactions were also analyzed for all tested variables. The final reduced model was obtained with the following predictor variable categories: age (45–64 and $\geq 65$ years) and smoking status (smoker). Odds ratio (OR) and 95% confidence intervals (95% CI) were calculated for both univariate and multivariate analyses. The level of statistical significance was set at 5%.

## RESULTS

Table 1 shows the distribution of sociodemographic, behavioral, biometric and oral hygiene data in the studied sample. Ages ranged from 20 to 90 years. The sample had 55.6% of

**Table 1  Sociodemographic, behavioural, biometric and oral hygiene data (N = 405).**

| Variable | | n (%) |
|---|---|---|
| Gender | Female | 225 (55.6) |
| | Male | 180 (44.4) |
| Age (years) | 20–44 | 90 (22.2) |
| | 45–64 | 217 (53.6) |
| | ≥65 | 98 (24.2) |
| Smoking status | Smoker | 141 (34.8) |
| | Non-smoker | 264 (65.2) |
| Active smokers (cigarettes per day) (n = 141) | Light (<10) | 41 (29.1) |
| | Medium (10–20) | 93 (66.0) |
| | Heavy (>20) | 7 (5.0) |
| Education | Elementary | 157 (38.8) |
| | Middle | 155 (38.3) |
| | Higher | 93 (23.0) |
| Employment status | Employed | 210 (51.9) |
| | Unemployed | 63 (15.6) |
| | Retired | 132 (32.6) |
| BMI (kg/m$^2$) | <18.5 | 5 (1.2) |
| | 18.5–24.9 | 162 (40.0) |
| | 25.0–29.9 | 159 (39.3) |
| | ≥30 | 79 (19.5) |
| Last dental visit | <1 year | 185 (45.7) |
| | 1–2 years | 57 (14.1) |
| | 3–4 years | 75 (18.5) |
| | ≥5 years | 83 (20.5) |
| | Never | 5 (1.2) |
| Consultation motive | Routine | 125 (30.9) |
| | Aesthetics | 35 (8.6) |
| | Pain | 73 (18.0) |
| | Functional | 157 (38.8) |
| | Other | 15 (3.7) |
| Dental floss usage | Yes | 141 (34.8) |
| | No | 264 (65.2) |
| Toothbrush frequency | 2–3 times/daily | 313 (77.3) |
| | 1 time/daily | 75 (18.5) |
| | 2–6 times/weekly | 17 (4.2) |

**Notes.**
BMI (kg/m$^2$), Body Mass Index (kilogram/meter$^2$).

female patients. It is worth to mention that 65.2% of subjects did not smoke and active smokers were mainly medium smokers (66%), followed by light smokers (29%) and heavy smokers (5%). Regarding education and employment status, 77.1% of subjects had elementary or middle education, and 51.9% of the subjects were employed. Approximately 59% were overweight and obese, and only 40% had normal values. Interestingly, 53.1% had

a period of over one year without any dental visit and 1.2% never had a dental appointment, whereas functional complaint was the major consultation motive.

Table 2 shows the periodontal data of this sample according to age, gender, and smoking status. Subjects over 65 years of age had a significantly higher mean number of missing teeth and, in total, this subpopulation presented a mean loss of 8 teeth. Younger individuals (<45 years of age) presented a significantly lower mean number of missing teeth, PD, REC, furcation lesions and teeth with mobility compared to older subjects. Male patients presented a significantly higher mean PD, deep periodontal pockets ($\geq$5 mm) and teeth with furcation lesions than female. Compared to smokers, non-smokers had lower mean SB, PD and CAL, and less deep periodontal pockets.

Chronic periodontitis was diagnosed in 83.5% of the patients (Table 3), and subjects with chronic periodontitis had CAL $\geq$3 mm, $\geq$4 mm, $\geq$5 mm, $\geq$6 mm and $\geq$7 mm affecting, on average, 83.7%, 54.4%, 32.1%, 17.8% and 9.2% of their teeth, respectively (Table 4). Besides, the first lower molar was the most frequently missing tooth, while the lower canine was the least lost but the most severely affected tooth (Fig. 1).

In the logistic regression analysis, similar results were observed in the univariable (Table 5) and multivariable models (Table 6). In the multivariable analysis, smoking (OR = 3.55, 95% CI [1.80–7.02]) and older age (OR = 8.70, 95% CI [3.66–20.69] and OR = 4.85, 95% CI [2.57–9.16]), for 65+ and 45–64 years old, respectively, were identified as risk indicators for CAL $\geq$ 3 mm (Table 5). Chronic periodontitis was not significantly associated with the remaining variables.

## DISCUSSION

This retrospective cross-sectional study assessed the periodontal status of forwarded adult subjects who sought dental treatment in a Portuguese university dental clinic, that is located in the metropolitan area of Lisbon. This area has over 2.8 million inhabitants and is the largest Portuguese metropolitan area (*Área Metropolitana de Lisboa, 2018*). This university dental clinic is an important reference dental center in the Lisbon Region and receives patients from all social strata. The absence of complete socioeconomic data constitutes a limitation of this study. Unfortunately, over 70% of patients (data not shown) refused to provide socioeconomic status information.

The results of this retrospective study can't be compared with previous investigations performed in Portugal because in these it was applied the CPITN methodology (*DGS, 2015*; *Freitas et al., 1983*; *Marques et al., 2000*; *Petersen & Ogawa, 2012*). This is the first FMPE protocol used in a Portuguese population and provides direct evidence for estimating periodontal status and results in a better representation of the population (*Eke et al., 2012b*). Although FMPE methodology can result in an overestimation of periodontal treatment needs among young adults (*Aimetti et al., 2015*), the partial-mouth examination can miscalculate the prevalence of periodontitis in almost 50% of the population (*Eke et al., 2012b*). The overall results demonstrate that this referred subpopulation had a high prevalence of chronic periodontitis (79.3%, 95% CI [77.5–88.1]%), and severe extensity of periodontal destruction among the affected subjects (83.7%, 95% CI [81.7–85.6]%).

Machado et al. (2018), *PeerJ*, DOI 10.7717/peerj.5258

**Table 2** Periodontal clinical data (presented as mean ± standard deviation) as a function of gender, age and smoking status ($N = 405$).

| | | SB (%) | BOP (%) | PD (mm) | REC (mm) | Missing teeth (n) | Teeth w/mobility (n) | Teeth w/furcation lesions (n) | Deep periodontal pockets (≥5 mm) (n) | CAL | | | |
|---|---|---|---|---|---|---|---|---|---|---|---|---|---|
| | | | | | | | | | | Total | ≥3 mm (%) | ≥5 mm (%) | ≥7 mm (%) |
| Gender | Female | 34.6 ± 22.8 | 9.1 ± 12.6 | 3.1 ± 0.7[*] | 1.0 ± 0.9 | 8.5 ± 5.9 | 5.2 ± 5.0 | 0.4 ± 0.8[*] | 15.0 ± 18.8[*] | 4.0 ± 1.2 | 77.6 ± 19.8[*] | 33.2 ± 24.8 | 11.1 ± 15.8[*] |
| | Male | 37.4 ± 23.6 | 11.3 ± 15.7 | 3.3 ± 0.8[*] | 1.0 ± 0.9 | 8.1 ± 5.6 | 4.4 ± 4.2 | 0.5 ± 0.9[*] | 20.1 ± 19.8[*] | 4.3 ± 1.5 | 81.5 ± 17.7[*] | 38.4 ± 27.1 | 14.8 ± 18.9[*] |
| Age (years) | 20–44 | 33.3 ± 20.9 | 9.9 ± 12.3 | 3.1 ± 0.7 | 0.6 ± 0.7[**] | 5.3 ± 5.0[**] | 4.0 ± 4.7[**] | 0.2 ± 0.5[**] | 18.4 ± 19.9 | 3.6 ± 1.2[**] | 72.5 ± 21.2[**] | 25.6 ± 24.3[**] | 7.6 ± 15.1[**] |
| | 45–64 | 35.4 ± 23.5 | 10.1 ± 14.6 | 3.3 ± 0.8 | 1.0 ± 0.9[**] | 8.5 ± 5.5[**] | 5.4 ± 5.0[**] | 0.5 ± 0.9[**] | 19.1 ± 21.2 | 4.3 ± 1.3[**] | 81.5 ± 18.2[**] | 38.5 ± 25.9[**] | 13.9 ± 17.4[**] |
| | ≥65 | 39.0 ± 24.1 | 10.1 ± 14.5 | 3.1 ± 0.7 | 1.2 ± 1.0[**] | 10.7 ± 6.0[**] | 4.5 ± 3.6[**] | 0.6 ± 0.9[**] | 12.3 ± 12.8 | 4.3 ± 1.4[**] | 80.8 ± 17.2[**] | 38.0 ± 25.5[**] | 14.7 ± 18.3[**] |
| Smoking status | Smoker | 38.7 ± 23.9[*] | 8.6 ± 14.2 | 3.4 ± 0.8[*] | 1.1 ± 1.0 | 8.2 ± 5.8 | 5.3 ± 5.2 | 0.4 ± 0.8 | 22.5 ± 22.0[*] | 4.5 ± 1.4[*] | 85.5 ± 16.1[*] | 42.9 ± 28.6[*] | 15.8 ± 19.6[*] |
| | Non-smoker | 34.3 ± 22.6[*] | 10.8 ± 13.9 | 3.1 ± 0.7[*] | 0.9 ± 0.8 | 8.4 ± 5.8 | 4.6 ± 4.3 | 0.5 ± 0.9 | 14.5 ± 17.3[*] | 3.9 ± 1.3[*] | 76.0 ± 19.6[*] | 31.6 ± 23.5[*] | 11.1 ± 15.8[*] |
| Total | | 35.8 ± 23.1 | 10.1 ± 14.1 | 3.2 ± 0.8 | 1.0 ± 0.9 | 8.3 ± 5.8 | 4.8 ± 4.6 | 0.4 ± 0.9 | 17.3 ± 19.4 | 4.1 ± 1.3 | 79.3 ± 19.0 | 79.3 ± 19.0 | 79.3 ± 19.0 |

**Notes.**

SB, Supragingival Biofilm; BOP, Bleeding on Probing; PD, Pocket Depth; REC, Recession; CAL, Clinical Attachment Loss.

[*]Mann–Whitney test ($p < 0.05$).

[**]Kruskal-Wallis test ($p < 0.05$).

Machado et al. (2018), *PeerJ*, DOI 10.7717/peerj.5258

**Table 3  Percentage of patients, with 95% confidence interval (95% CI), by threshold of CAL (mm), severity and age group (years).**

| CAL (mm) | Subjects with chronic periodontitis | | | | | | | | All subjects | | | | | | | |
|---|---|---|---|---|---|---|---|---|---|---|---|---|---|---|---|---|
| | 20–44 (n = 59) | | 45–64 (n = 190) | | ≥65 (n = 89) | | Total (n = 338) | | 20–44 (n = 90) | | 45–64 (n = 217) | | ≥65 (n = 98) | | Total (n = 405) | |
| | % | 95% CI | % | 95% CI | % | 95% CI | % | 95% CI | % | 95% CI | % | 95% CI | % | 95% CI | % | 95% CI |
| **Prevalence (patients)** | | | | | | | | | | | | | | | | |
| ≥3 | 100 | 100.0–100.0 | 100 | 100.0–100.0 | 100 | 100.0–100.0 | 100 | 100.0–100.0 | 65.6 | 56.1–75.1 | 87.6 | 83.5–91.7 | 90.8 | 85.3–96.3 | 83.5 | 80.4–86.6 |
| ≥4 | 42.4 | 30.0–54.8 | 62.1 | 55.7–68.6 | 59.6 | 49.7–69.5 | 58.0 | 53.4–62.6 | 27.8 | 18.8–36.8 | 54.4 | 48.3–60.5 | 54.1 | 44.6–63.6 | 48.4 | 44.2–52.6 |
| ≥5 | 20.3 | 10.2–30.4 | 30.0 | 23.9–36.1 | 25.8 | 17.0–34.6 | 15.4 | 12.0–18.8 | 13.3 | 6.5–20.1 | 26.3 | 20.9–31.7 | 23.5 | 15.4–31.6 | 12.8 | 10.0–15.6 |
| ≥6 | 8.5 | 1.515.5 | 13.2 | 8.7–17.7 | 12.4 | 5.8–19.0 | 12.1 | 9.0–15.2 | 5.6 | 1.0–10.2 | 11.5 | 7.6–15.4 | 11.2 | 5.2–17.2 | 10.1 | 7.6–12.6 |
| ≥7 | 5.1 | 0.0–10.6 | 0.5 | 0.0–1.4 | 5.6 | 1.0–10.2 | 5.0 | 3.0–7.1 | 3.3 | 0.0–6.9 | 4.1 | 1.7–6.5 | 5.1 | 0.9–9.3 | 4.2 | 2.5–5.9 |

**Notes.**

CI,  Confidence Interval;  CAL,  Clinical Attachment Loss.

Machado et al. (2018), *PeerJ*, DOI 10.7717/peerj.5258

**Table 4  Percentage with 95% confidence interval (95% CI), of sites (prevalence) and affected teeth (extent), by threshold of CAL (mm), severity and age group (years).**

| CAL (mm) | Subjects with chronic periodontitis | | | | | | | | All subjects | | | | | | | |
|---|---|---|---|---|---|---|---|---|---|---|---|---|---|---|---|---|
| | 20–44 (*n* = 59) | | 45–64 (*n* = 190) | | ≥65 (*n* = 89) | | Total (*n* = 338) | | 20–44 (*n* = 90) | | 45–64 (*n* = 217) | | ≥65 (*n* = 98) | | Total (*n* = 405) | |
| | % | 95% CI | % | 95% CI | % | 95% CI | % | 95% CI | % | 95% CI | % | 95% CI | % | 95% CI | % | 95% CI |
| **Prevalence (sites)** | | | | | | | | | | | | | | | | |
| ≥3 | 85.3 | 82.7–87.4 | 86.9 | 85.3–88.5 | 84.4 | 81.7–87.1 | 86.0 | 84.7–87.2 | 72.5 | 68.0–76.9 | 81.5 | 79.0–83.9 | 80.8 | 77.4–84.2 | 79.3 | 77.5–81.2 |
| ≥4 | 57.0 | 51.6–62.4 | 61.3 | 58.1–64.5 | 59.2 | 54.5–63.9 | 60.0 | 57.6–62.4 | 41.5 | 35.8–47.1 | 55.2 | 51.7–58.7 | 55.3 | 50.2–60.4 | 52.2 | 49.6–54.8 |
| ≥5 | 36.9 | 30.7–43.0 | 43.3 | 39.8–46.8 | 41.2 | 36.1–46.3 | 41.6 | 39.0–44.2 | 25.6 | 20.5–30.7 | 38.5 | 35.1–42.0 | 38.0 | 33.0–43.1 | 35.5 | 33.0–38.1 |
| ≥6 | 21.9 | 16.4–27.4 | 27.7 | 24.5–30.9 | 26.8 | 22.2–31.4 | 26.5 | 24.1–28.9 | 14.7 | 10.6–18.8 | 24.5 | 21.5–27.5 | 24.6 | 20.1–29.1 | 22.4 | 20.2–24.5 |
| ≥7 | 11.7 | 7.0–16.4 | 15.8 | 13.2–18.3 | 16.1 | 12.2–20.1 | 15.2 | 13.2–17.1 | 7.6 | 4.4–10.8 | 13.9 | 11.6–16.2 | 14.7 | 11.0–18.4 | 12.7 | 11.0–14.4 |
| **Extent (affected teeth)** | | | | | | | | | | | | | | | | |
| ≥3 | 82.1 | 77.6–86.6 | 85.0 | 82.5–87.7 | 81.7 | 77.6–85.7 | 83.7 | 81.7–85.6 | 62.1 | 55.5–68.7 | 77.3 | 73.7–80.9 | 77.0 | 72.3–81.7 | 73.9 | 71.1–76.6 |
| ≥4 | 49.3 | 41.9–56.7 | 56.1 | 51.9–60.3 | 54.0 | 47.8–60.2 | 54.4 | 51.3–57.5 | 33.5 | 27.1–39.9 | 49.8 | 45.5–54.0 | 49.5 | 43.2–55.8 | 46.1 | 43.0–49.2 |
| ≥5 | 25.0 | 17.8–32.1 | 33.4 | 29.2–37.6 | 33.8 | 27.8–39.8 | 32.1 | 29.0–35.2 | 16.0 | 10.9–21.2 | 29.5 | 25.6–33.4 | 30.8 | 25.0–36.5 | 26.8 | 24.0–29.6 |
| ≥6 | 12.3 | 6.8–17.8 | 18.6 | 15.2–22.0 | 19.5 | 14.3–24.7 | 17.8 | 15.2–20.3 | 7.9 | 4.2–11.6 | 16.3 | 13.2–19.3 | 17.7 | 12.8–22.6 | 14.7 | 12.6–16.9 |
| ≥7 | 5.8 | 2.1–9.4 | 10.0 | 7.5–12.4 | 9.9 | 5.8–14.1 | 9.2 | 7.4–11.1 | 3.7 | 1.3–6.0 | 8.7 | 6.5–10.8 | 9.0 | 5.2–12.8 | 7.6 | 6.1–9.2 |

**Notes.**

CI, Confidence Interval; CAL, Clinical Attachment Loss.

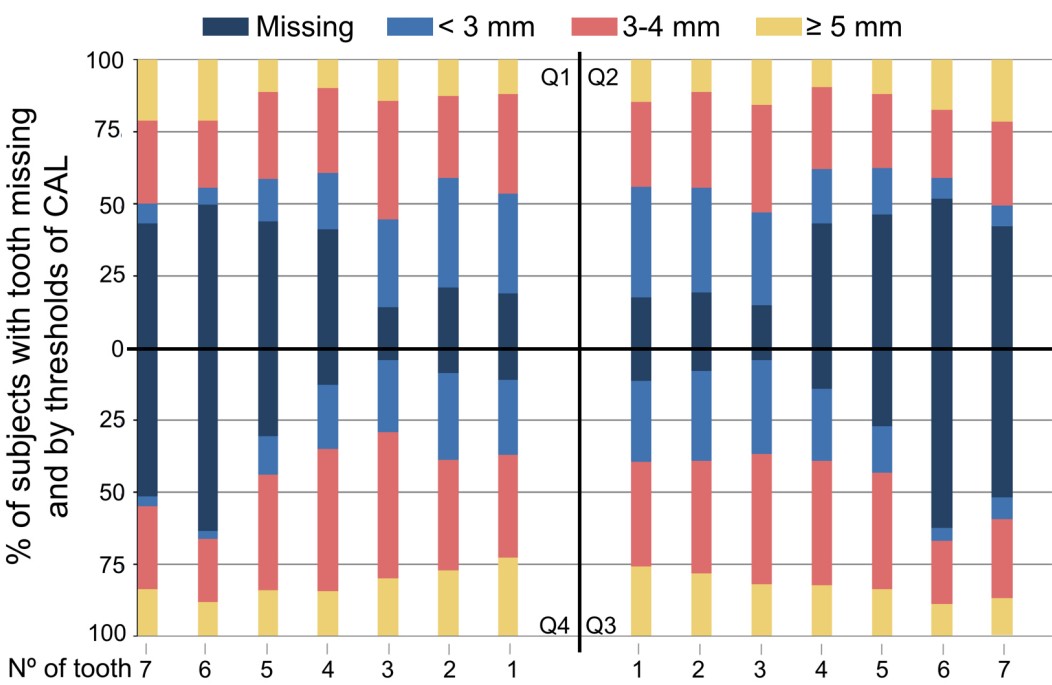

**Figure 1** **Percentage of subjects with the respective tooth present and by thresholds of CAL (mm), at each specific position, for all teeth in all quadrants.** The black lines indicate the separation by each quadrant. Dark blue, percentage of missing teeth; Blue, percentage of teeth with less than 3 mm of CAL; pink, percentage of teeth with 3–4 mm of CAL; yellow, percentage of teeth over 4 mm of CAL.

This investigation study design is not an epidemiological study per se, but rather an observational study of patients who were forwarded to a periodontology consultation. Thus, we were only able to estimate the prevalence and extent of our referred subpopulation. However, these results underline the fact that the majority of patients attended the periodontal consultation already in a state of advanced periodontal destruction and only a small percentage appeared in the early stages or healthy. Still, a disturbing percentage of patients did not attend periodontal consultations despite the triage referral with approximately 69% missing or unchecking the appointment.

Regarding tooth loss, the most frequently missed teeth were the lower first molars and the less missed were the lower canines, as with recent European data (*Aimetti et al., 2015*; *Schutzhold et al., 2015*). Additionally, lower canines and incisors were the most affected teeth with CAL and the lower molars the less. The lower arch presented more periodontal destruction than the upper, and the teeth with more severe CAL levels in the upper arch were the canines.

Concerning periodontal parameters, unlike PD, CAL severity increased with age and can be related to the increase of gingival recession with aging (*Eke et al., 2012a*). As in the literature (*Bouchard et al., 2006*; *Bourgeois, Bouchard & Epidemiology, 2007*; *Aimetti et al., 2015*; *Schutzhold et al., 2015*; *Holde et al., 2017*), age was confirmed in the multivariate analysis as a risk indicator for chronic periodontitis for 45–64 years old (OR = 4.85, 95% CI [2.57–9.16]) and 65+ years old (OR = 8.70, 95% CI [3.66–20.69]). However, it is

**Table 5** Univariate logistic regression analysis of sociodemographic, behavioural, anthropometric and oral hygiene variables for the outcome variable CAL $\geq$ 3 mm ($N = 405$).

| Predictor variables | | OR (95% CI) | p |
|---|---|---|---|
| Gender | Female | 1 | – |
| | Male | 1.32 (0.77–2.26) | 0.310 |
| | | | <0.001 |
| Age (years) | 20–44 | 1 | – |
| | 45–64 | 3.70 (2.04–6.69) | <0.001 |
| | ≥65 | 5.20 (2.31–11.70) | <0.001 |
| Smoking status | Smoker | 2.06 (1.11–3.81) | 0.021 |
| | Non-smoker | 1 | – |
| | | | 0.107 |
| Education (years) | 1–4 | 1.40 (0.74–2.66) | 0.298 |
| | 5–12 | 2.09 (1.05–4.13) | 0.035 |
| | >12 | 1 | – |
| | | | 0.246 |
| Employment status | Employed | 1 | – |
| | Unemployed | 1.67 (0.74–3.77) | 0.219 |
| | Retired | 1.54 (0.84–2.81) | 0.163 |
| | | | 0.699 |
| BMI (kg/m$^2$) | <18.5 | 1 | – |
| | 18.5–24.9 | 1.06 (0.11–9.79) | 0.961 |
| | 25.0–29.9 | 1.48 (0.16–13.82) | 0.732 |
| | ≥30 | 1.40 (0.14–13.59) | 0.774 |
| | | | 0.026 |
| Last dental visit | <1 year | 1 | – |
| | 1–2 years | 1.39 (0.54–3.57) | 0.493 |
| | 3–4 years | 0.42 (0.22–0.81) | 0.009 |
| | ≥5 years | 0.97 (0.46–2.03) | 0.930 |
| | Never | 0.24 (0.04–1.54) | 0.134 |
| | | | 0.806 |
| Consultation motive | Routine | 1 | – |
| | Aesthetics | 0.72 (0.29–1.80) | 0.483 |
| | Pain | 1.09 (0.50–2.35) | 0.834 |
| | Functional | 1.24 (0.66–2.36) | 0.502 |
| | Other | 1.39 (0.29–6.60) | 0.680 |
| Dental floss use | Yes | 1 | – |
| | No | 1.66 (0.97–2.82) | 0.063 |
| | | | 0.803 |
| Toothbrush frequency | 2–3 times/daily | 1 | – |
| | 1 time/daily | 1.27 (0.63–2.56) | 0.508 |
| | 2–6 times/weekly | – | 0.998 |

**Notes.**
BMI (kg/m$^2$), Body Mass Index (kilogram/meter$^2$); CI, Confidence Interval; OR, Odds Ratio.

**Table 6  Multivariate logistic regression analysis (final reduced model) (*) for the outcome variable CAL ≥ 3 mm (N = 405).**

| Predictor variables | | CAL ≥ 3 mm | |
| --- | --- | --- | --- |
| | | OR (95% CI) | $p$ |
| Age (years) | 20–44 | 1 | – |
| | 45–64 | 4.85 (2.57–9.16) | <0.001 |
| | ≥65 | 8.70 (3.66–20.69) | <0.001 |
| Smoking status | Non-smoker | 1 | – |
| | Smoker | 3.55 (1.80–7.02) | <0.001 |

Notes.

CI, Confidence Interval; OR, Odds Ratio; CAL, Clinical Attachment Loss.
(*) The model was statistically significant, $\chi^2(3) = 39.507$, $p < 0.001$, explained 15.7% (Nagelkerke $R^2$) of the variance and correctly classified 83.5% of cases.

important to highlight that, in the majority of CAL thresholds of subjects with the disease, the 45–64 years old group presented worse results for prevalence of chronic periodontitis, while 65+ years old group had worse levels of periodontal destruction extent.

Smoking was strongly associated with chronic periodontitis (OR = 3.55, 95% CI [1.80–7.02]). Previous studies reported OR values ranging between 2 and 9 of having periodontitis (*Aimetti et al., 2015*; *Schutzhold et al., 2015*; *Holde et al., 2017*; *Bergström & Preber, 1994*; *Bergström, 2006*; *Kinane, Stathopoulou & Papapanou, 2017*; *Silva-Boghossian, Luiz & Colombo, 2009*). Despite not accounting for lifetime smoking exposure, we stratified current smokers according to the number of cigarettes smoked although it was not significantly associated with the severity and progression of the periodontal disease.

Several studies found that obesity was associated with an increased risk of periodontitis (*Nishida et al., 2005*; *Saito et al., 2005*; *Dalla Vecchia et al., 2005*). Besides that, *Suvan et al. (2015)* concluded that overweight/obese individuals are more likely to suffer from periodontitis compared to normal weight individuals. Although our results show that overweight and obesity have no impact on the aggravation of periodontitis, we emphasize that more than half of this subpopulation was overweight or obese, in agreement with the latest national IAN-AF Food and Activity Survey (*IAN-AF, 2016*).

In the past, several epidemiological surveys reported that people with lower educational level had higher prevalence and severity of periodontal disease (*Bourgeois, Bouchard & Epidemiology, 2007*; *Aimetti et al., 2015*; *Holde et al., 2017*; *Albandar, Brunelle & Kingman, 1999*; *Krustrup & Erik Petersen, 2006*). However, other studies have indicated that this impact cannot be seen in a singular way but in a multifactorial view (*Albandar, 2002*; *Geyer, Schneller & Micheelis, 2010*). Our results show that despite middle education had significance in the univariable model (OR = 2.09 (95% CI [1.05–4.13]), $p = 0.035$), when analyzed in a multivariable model it had no impact on the probability of having chronic periodontitis.

## CONCLUSION

This specific subpopulation of individuals referred to periodontal examination in a university dental clinic of the Lisbon region presented high prevalence and severe extent

of chronic periodontitis. Age and smoking were identified as risk indicators for chronic periodontitis in this referred subpopulation. Within the limitations of this study, these results highlight the importance of developing appropriate public health programs to educate the Portuguese population about the burden of periodontal diseases.

## ACKNOWLEDGEMENTS

We thank Júlio Guilherme for his substantial support for database development and management during this study.

### Funding

The authors received no funding for this work.

### Competing Interests

The authors declare there are no competing interests.

### Author Contributions

- Vanessa Machado and João Botelho conceived and designed the experiments, performed the experiments, analyzed the data, contributed reagents/materials/analysis tools, prepared figures and/or tables, authored or reviewed drafts of the paper, approved the final draft.
- António Amaral and Ricardo Alves performed the experiments, analyzed the data, authored or reviewed drafts of the paper, approved the final draft.
- Luís Proença conceived and designed the experiments, analyzed the data, prepared figures and/or tables, authored or reviewed drafts of the paper, approved the final draft.
- João Rua, Ana Sintra Delgado and José João Mendes analyzed the data, contributed reagents/materials/analysis tools, authored or reviewed drafts of the paper, approved the final draft.
- Maria Alzira Cavacas analyzed the data, authored or reviewed drafts of the paper, approved the final draft.

### Human Ethics

The following information was supplied relating to ethical approvals (i.e., approving body and any reference numbers):

The study was conducted in accordance with the Declaration of Helsinki of 1975, as revised in 2013, and approved by the Ethics Committee of Egas Moniz (Ethical Application Ref: 595).

### Data Availability

João Botelho. (2018). Periodontal data portuguese population (Version 1) [Data set]. Zenodo. http://doi.org/10.5281/zenodo.1173340.

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
