# Peer review of "Prevalence and extent of chronic periodontitis and its risk factors in a Portuguese subpopulation: a retrospective cross-sectional study and analysis of Clinical Attachment Loss"

_PeerJ, doi:10.7717/peerj.5258_

## Round 0.1 · original submission · Major Revisions

Dear Author,

Please address all of the reviewers comments, in particular the comments from Reviewer 3 need some serious attention. In addition, you must improve the English language - we suggest you seek an English editing service, and provide their certificate.

Reviewer 1 ·

Basic reporting

Reviewers Comments:
Abstract: Very well written. All aspects of the study have been covered nicely.
Introduction: Relative and contemporary.

Experimental design

M & M: A convenient sample from a university clinical facility. Clinical parameters and health questionnaire, well explained. Inter-examiner correlation presented. Very appropriate use of statistical methods.

Validity of the findings

Results: Well explained.
Discussion: Compared with other studies and provided the rationale of the study design etc.
Conclusion: The prevalence and severity of the subpopulation used in this study gave upward results. Risk indicators are comparable with other studies from international populations.

Additional comments

Recommendation: In future, a true representative sampling technique should be used. The comparison of FMPE and partial recording of the same population, using CPITN should be used to find out over or underestimation of the periodontal disease. The findings of the current examined population cannot be generalized to general population of the city or country.
Final Recommendation:

Reviewer 2 ·

Basic reporting

No comment

Experimental design

A question regarding selection of the subpopulation:

From reading the materials and methods, my understanding is selecting patients who were specifically referred to the periodontology department to obtain a periodontal examination, after being screened in triage. Therefore, all cases that are referred are suspected to have periodontal disease. As a result, the prevalence will accordingly be high. Why not calculate the prevalence of the original population that was screened from your target period: September 2015 to March 2017, which was a total of 3648 patients. This would give us a better idea on the prevalence in this subpopulation.

Validity of the findings

Regarding the following sentence in the discussion (lines 211-213):

“However, these results underline the fact that the majority of patients attended the periodontal consultation already in a state of advanced periodontal destruction and only a small percentage appeared in the early stages or healthy”

Can you define the threshold where triage sends the patients to the periodontal department? Their threshold could be high, and therefore maybe missing the mild periodontitis cases.
Maybe you can expand on this point in your manuscript.

Reviewer 3 ·

Basic reporting

The Authors should clarify the fooling questions:
• The initials (name) of the calibrated periodontists;
• Bibliography references formatting –“…. Four BMI categories were defined using WHO criteria (Kopelman 2000)…”
• Bibliography reference to support this statement is missing: “This area has over 2.8 million inhabitants and is the largest Portuguese metropolitan area”
• All tables need to have legend of the acronyms

Finally, it is advisable to have the text revised by a native English speaker since it would help the text understanding, there are some misspelling and grammatical issues.
For instance, “STrengthening the Reporting of OBservational studies in Epidemiology (STROBE)” or “27% of the patients indicated for periodontal treatment”

Experimental design

. I felt a bit lost: is this a prospective or retrospective study? “Study was carried out between September 2015 and March 2017”, but when we analyze the documents, the ethic commission only gave the approval on 29th of November 2017, after the study was performed. In any part of the paper is mentioned that this was a retrospective cross-sectional study.

. The informed consent doesn’t respect the Declaration of Helsinki revised in 2013 where: “…In medical research involving human subjects capable of giving informed consent, each potential subject must be adequately informed of the aims, methods, sources of funding, any possible conflicts of interest, institutional affiliations of the researcher, the anticipated benefits and potential risks of the study and the discomfort it may entail, post-study provisions and any other relevant aspects of the study. The potential subject must be informed of the right to refuse to participate in the study or to withdraw consent to participate at any time without reprisal. Special attention should be given to the specific information needs of individual potential subjects as well as to the methods used to deliver the information.” Analyzing the informed consent attached to the paper, even the study name is missing among with most of the others requirements of Helsinki.

. How it was performed the first triage to select the patients to be treated into the periodontology department?

Validity of the findings

. The purpose of this study was “to assess the prevalence, severity and extent of chronic periodontitis through the full-mouth examination of CAL, and its association with sociodemographic, behavioral and environmental risk factors, in a Portuguese adult subpopulation of a suburban area of the Lisbon Region forwarded to periodontal examination”. The authors report that the “Prevalence of chronic periodontitis was 83.5%”.

• The authors only included in the study a pre-selected sample that was referred to the periodontology department. Due to that, is quite normal that the percentage of chronic periodontitis was very high. From my point of view, is absolutely necessary to explain that result tacking in consideration the sample selected, this result can’t be reported as absolute. [“From a total of 3648 subjects who sought a first consultation in the university dental clinic during that period, 1501 (41%) patients were referred to the periodontology department, based on their triage status. From these, 459 attended a periodontal consultation and were considered for this study.”]

• For the same reason, most of the conclusion aren’t accord of the performed study eg. “Age and smoking were identified as risk indicators for chronic periodontitis” because this conclusion are based only in “patients that have been referred to the periodontology department based on their triage status”, so is normal most of them have periodontitis.

Additional comments

This cross-sectional study, as is stated in the conclusions, is a good base to “prepare future detailed epidemiological studies and appropriate public health programs”.
However, further clarifications are required to make this cross-sectional study suitable for publication.

---

## Round 0.2 · Minor Revisions

Please address the reviewer comments in a revised submission.

Reviewer 3 ·

Basic reporting

the authors improved the paper and answered questions.

Experimental design

Not only in the line89 but all the paper including the title should be written that this is a retrospective cross-sectional.
The report at eg. line 177, 207, 240 are statements of a subpopulation not a population.
The autores need to review this statement along all the paper.

Validity of the findings

the authors improved the paper and answered questions.

Additional comments

Not only in the line89 but all the paper including the title should be written that this is a retrospective cross-sectional.
The report at eg. line 177, 207, 240 are statements of a subpopulation not a population.
The autores need to review this statement along all the paper.

---

## Round 0.3 · accepted · Accept

Congratulations, your article is now Accepted.